# Enhanced Electrochemical Performance of LiNi$_{1/3}$Co$_{1/3}$Mn$_{1/3}$O$_2$ at a High Cut-Off Voltage of 4.6 V by Li$_{1.3}$Al$_{0.3}$Ti$_{1.7}$(PO$_4$)$_3$ Coating

**Ming Zhang** [1,2]**, Peng Zhang** [1,*]**, Weidong Wen** [1]**, Huanwen Wang** [1]**, Beibei He** [1]**, Yansheng Gong** [1]**, Jun Jin** [1] **and Rui Wang** [1,*]

---

[1] Faculty of Materials Science and Chemistry, China University of Geosciences, Wuhan 430074, China
[2] Wuhan Institute of Marine Electric Propulsion, Wuhan 430064, China
* Correspondence: zp@cug.edu.cn (P.Z.); wangrui@cug.edu.cn (R.W.)

**Abstract:** At present, LiNi$_{1/3}$Co$_{1/3}$Mn$_{1/3}$O$_2$ (NCM) is a widely used material in the commercial market due to the easy control of the preparation process and usage environment. However, its capacity keeps fading when the cut-off voltage increases. In this research, an Li$_{1.3}$Al$_{0.3}$Ti$_{1.7}$(PO$_4$)$_3$ (LATP) coating method is proposed to improve the cycle performance of LiNi$_{1/3}$Co$_{1/3}$Mn$_{1/3}$O$_2$ at a high cut-off voltage of 4.6 V. The battery prepared with LATP-modified NCM exhibits an increased discharge capacity retention of 92.37% after 100 cycles at 0.2C (1C = 200 mA g$^{-1}$), while the bare NCM only presents 64.28%. Our results indicate that LATP-surface coating might be a useful method to increase the cycle stability of NCM and other high-capacity cathode materials.

**Keywords:** lithium-ion batteries; cathode; LiNi$_{1/3}$Co$_{1/3}$Mn$_{1/3}$O$_2$; Li$_{1.3}$Al$_{0.3}$Ti$_{1.7}$(PO$_4$)$_3$; surface modification





## 1. Introduction

Recently, the dramatic development of the automobile industry has put forward higher requirements for batteries with high specific energy densities. Cathode is an important component of lithium-ion batteries, and improving its electrochemical performance is one of the hot spots at present [1–3]. Typical cathode materials include α-NaFeO$_2$-type LiCoO$_2$ and LiNiO$_2$, spinel LiMn$_2$O$_4$, and olivine LiFePO$_4$ [4]. Among them, LiCoO$_2$ is the first commercial cathode material, but the high cost of Co elements limits its applications in new electric vehicles [5–7]. In 1999, based on the study of layered binary cathode materials, Liu et al. first proposed to replace Ni in LiNiO$_2$ with Co and Mn to prepare LiNi$_{1-x-y}$Co$_x$Mn$_y$O$_2$ cathode materials [8]. This kind of material shows some advantages, such as low cost high specific capacity. Therefore, this type of ternary material is considered to be an important material to substitute commercial LiCoO$_2$ [9–12]. Currently, the elemental ratios of ternary materials used in industry are 111, 523, 622, and 811, among which NCM111(NCM) is a widely used material in the industry due to the easy control of the preparation process and usage environment [13,14]. However, compared with other ternary materials, the energy density of NCM material is still unsatisfactory and needs to be improved [15,16]. The easiest way to do so is to increase the cut-off voltage. Unfortunately, when the cut-off voltage reaches a certain value, the cathode electrolyte interphase (CEI) becomes unstable, and it continuously decomposes and regenerates in the charge and discharge cycles [17–20]. This continuous side reactions leads to increased internal resistance, which leads to poor battery performance.

Surface modification is a widely used and effective way to alleviate the surface side reactions of cathodes [21,22]. Huang et al. employed a one-step method and successfully coated carbon on the NCM surface [23]. This surface layer effectively enhanced the electron conductivity and inhibited the occurrence of side reactions between the cathode and electrolyte. Similarly, Wang et al. reported coating AlPO$_4$ on NCM by a chemical deposition method and improved the electrochemical performance [24]. Besides, several metal oxide

layers have been reported to improve the performance of NCM, such as $Li_2MoO_4$ [25], $B_2O_3$ [26], $Al_2O_3$ [27], $Sb_2O_3$ [28], and $V_2O_5$ [29], and they all show promising effects.

$Li_{1.3}Al_{0.3}Ti_{1.7}(PO_4)_3$ (LATP) is a lithium ionic conductor and considered to be a promising solid-state electrolyte in lithium batteries [30–32]. It is reported to be an effective layer on a $LiCoO_2$ surface to increase the electrochemical performance. The reason may be that it could protect the surface from the corrosion by HF, which is generated from the electrolyte decomposition at a high cut-off voltage of 4.6V [33,34]. Layered Li-Ni-Co-Mn-O based material is also a widely applied commercial material. Till now, an LATP layer has been used as a surface layer in NCM811 [35] and NCM622 materials [36]. In this case, the effects of an LATP layer on the performance of NCM111 material is a worthy question to study. In this work, LATP was employed on the surface of NCM to alleviate surface side reactions. The LATP layer was prepared with a simple wet chemical method. It was found that the LATP-modified NCM exhibits an increased discharge capacity retention of 92.37% after 100 cycles at 0.2C, while the bare NCM only presents 64.28%. This result indicates that LATP-surface coating might be an effective method to increase the cycle stability of NCM at a high cut-off voltage of 4.6 V, which would improve the energy density of NCM in practical usage.

## 2. Experimental

### 2.1. Preparation of Materials

Commercial $LiNi_{1/3}Co_{1/3}Mn_{1/3}O_2$ (NCM) powder was used in this research. The $Li_{1.3}Al_{0.3}Ti_{1.7}(PO_4)_3$-modified NCM samples (NCM@LATP) were synthesized by a wet chemical method, and the schematic diagram is shown in Figure 1. The precursors were $LiNO_3$ (99%,), $AlNO_3 \cdot 9H_2O$ (AR), $C_{12}H_{28}O_4Ti$ (98%), and $NH_4H_2PO_4$ (98%). Typically, stoichiometric amounts of precursors were dissolved in a beaker, then commercial NCM was added in. The weight ratio of LATP was set to be 0.5 wt% (NCM@0.5LATP) and 1 wt% (NCM@1LATP), respectively. The mixture was heated and stirred for 8 h. When the solution was about to evaporate to dryness, it was put into an oven and dried at 60 °C to completely evaporate the remaining solvent. The collected powder was pre-heated at 300 °C for 3 h, and then sintered at 800 °C for 2 h to obtain the final sample.

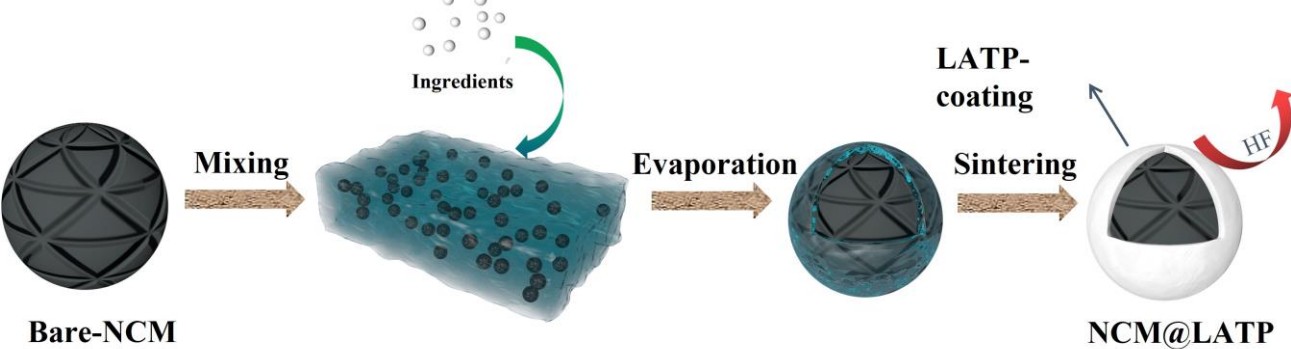

**Figure 1.** Schematic illustration for the LATP coating process.

### 2.2. Characterization of Materials

Crystalline structures of the samples were identified by X-ray diffraction (XRD, D8 Advance, Bruker, Karlsruhe, Germany). Scanning electron microscopy (SEM, Hitachi SU3500, Hitachi, Tokyo, Japan) was performed to check the morphology of the samples. Detailed morphological and lattice information were tested using a transmission electron microscopy instrument (TEM, Tecnai G2 F20, FEI, Eindhoven, The Netherland). X-ray photoelectron spectroscope (XPS, Thermo Scientific K-Alpha, Waltham, MA, USA) was used to detect the surface composition. Surface area analyzer (ASAP2460, Micromeritics, Norcross, GA, USA) was used to obtain the $N_2$ adsorption/desorption isotherms.

*2.3. Electrochemical Characterization*

Electrochemical characterization was performed following earlier reports [34]. The cells were cycled between 2.5 and 4.6 V at a rate of 0.2C or 1C (1C = 200 mA g$^{-1}$).

## 3. Results and Discussion

Figure 2 shows the XRD patterns of the two modified materials with different ratios (NCM@0.5LATP and NCM@1LATP) and the untreated material (bare NCM). All the diffraction peaks can be ascribed to the layered hexagonal α-NaFeO$_2$ type structure. The two obvious split peaks from (006)/(012) and (018)/(110) also infer that all three sample presents typical layered structures [37]. This means the structure of NCM remained unchanged after the treatment. According to the XRD results, no diffraction peaks of LATP were observed.

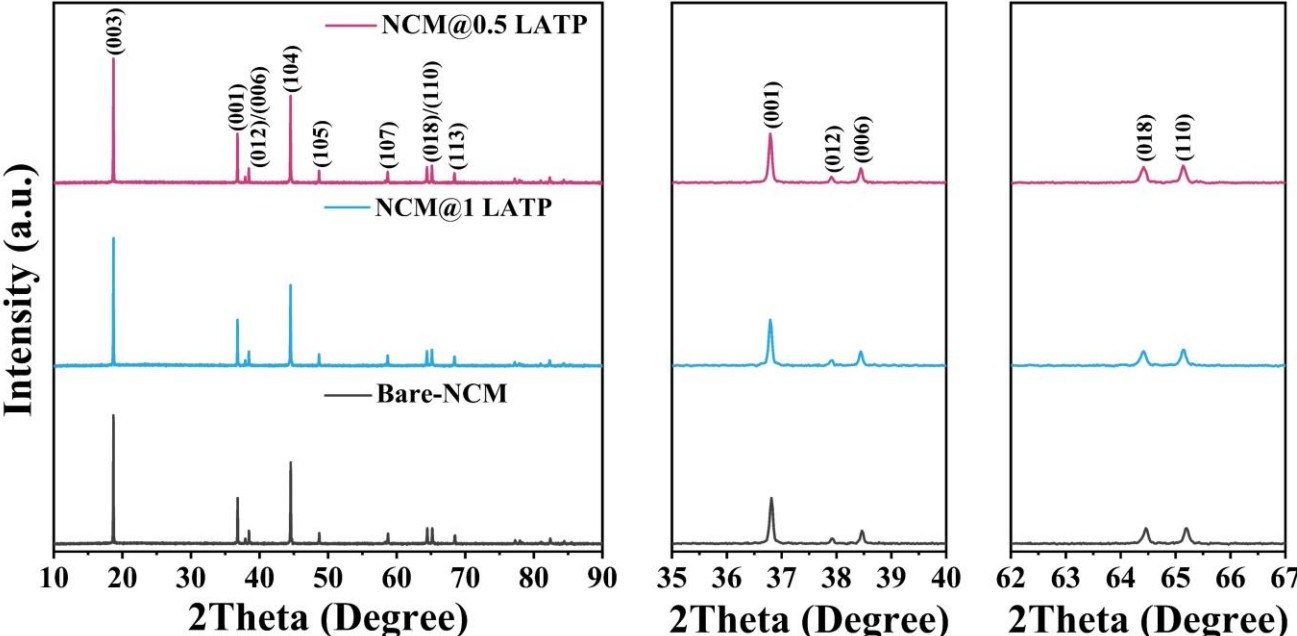

**Figure 2.** XRD patterns of bare NCM, NCM@0.5LATP, and NCM@1LATP.

Figure 3a–c show the SEM images of the three samples. According to the results, the bare NCM is basically uniform in size with a micron-level structure. The particles are in the shape of small spheres with good crystallization. Due to the relatively large size of the particles, it is difficult to see whether the coating is successful based on the particle morphology alone. In this case, NCM@0.5LATP was chosen for further study. EDS result confirms the existence of P, Al, and Ti elements in this sample. Elemental mapping results are shown in Figure 3e–k. It can be seen that the elements of Ni, Co, and Mn are distributed uniformly in the NCM@0.5LATP, and the shape of spherical particles can be clearly observed. By comparison, for the results of P, Al, and Ti, the spherical contour is not obvious. The reason may be that the coating layer is very thin and the element contents of P, Ti, and Al are very low. These results indicate that LTAP exists on the surface of NCM particles.

To further demonstrate the existence of the LATP layer more intuitively, the surfaces of the bare NCM and NCM@0.5LATP samples were inspected by TEM. As illustrated in the TEM images shown in Figure 4a,b, clear lattice fringes can be observed up to the edge. The interplanar distance was calculated to be 0.23 nm, and it should be related to the (003) crystal plane. For the NCM@0.5LATP sample, an amorphous layer can be seen on the surface. This should be the LATP layer, which indicates LATP might coat on the surface in the amorphous form.

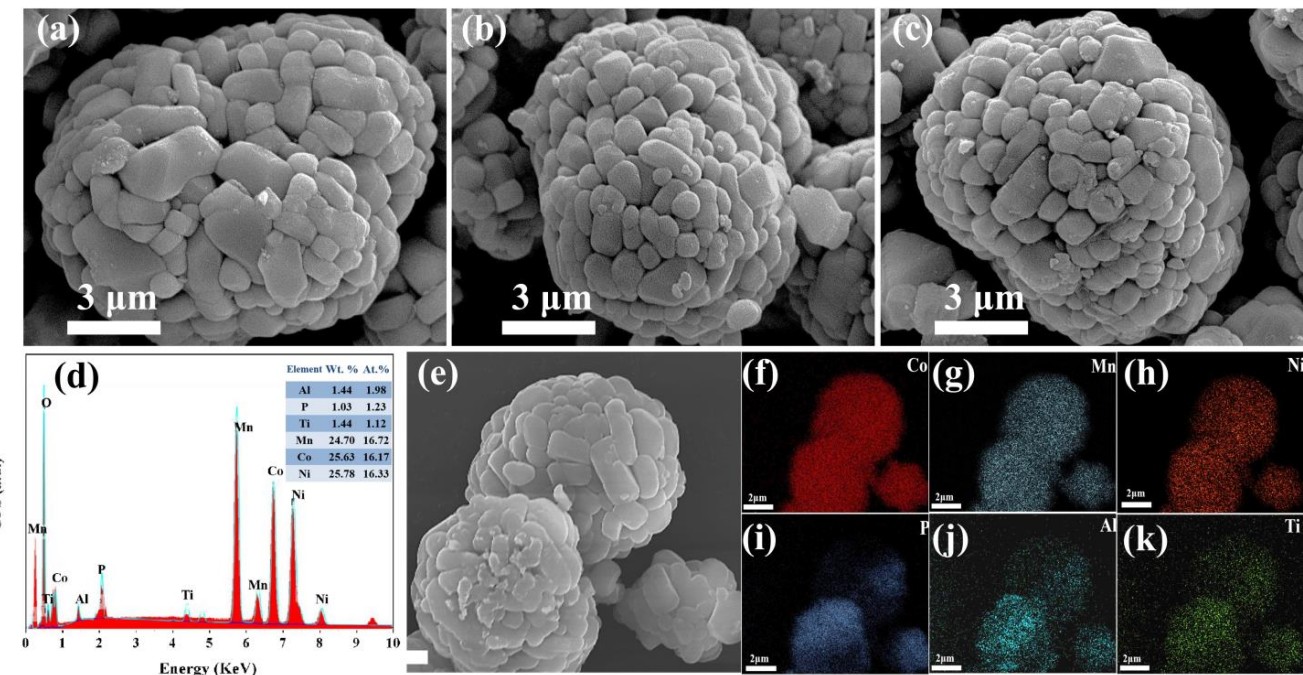

**Figure 3.** SEM images of (**a**) bare NCM, (**b**) NCM@0.5LATP, and (**c**) NCM@1LATP; (**d**) EDS spectrum; (**e**) area for the elemental mapping; elemental mapping of Co (**f**), Mn (**g**), Ni (**h**), P (**i**), Al (**j**), and Ti (**k**).

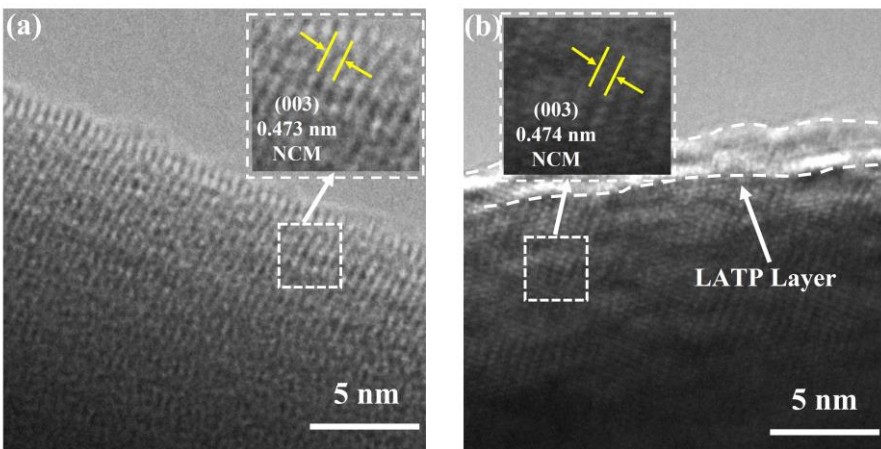

**Figure 4.** TEM images of (**a**) bare NCM and (**b**) NCM@0.5LATP.

Surface compositions of bare NCM and NCM@0.5LATP were analyzed with XPS. Firstly, the presence of characteristic peaks of Ni, Co, O, Mn, and C elements can be observed from the full spectrum of bare NCM, which is consistent with the previous EDS results. While for the NCM@0.5LATP sample, extra characteristic peaks of Al, Ti, and P are also observed in the spectrum. This result indicates that Al, Ti and P exist on the surface of NCM@0.5LATP. Figure 5b,c show the high-resolution XPS spectra of the P and Ti elements in the NCM@0.5LATP sample. Peaks at 458.3 eV and 463.9 eV can be ascribed to the binding energies of Ti $2p_{3/2}$ and Ti $2p_{1/2}$, which correspond to Ti$^{4+}$ ions in the Ti-based compounds [38]. Figure 5d–f show the high-resolution XPS spectra of the Co, Mn, and Ni elements. The peaks of the three elements are nearly the same in the bare NCM and NCM@0.5LATP samples, which indicates the valence states of Ni, Co, and Mn remained unchanged after the LATP modification [39].

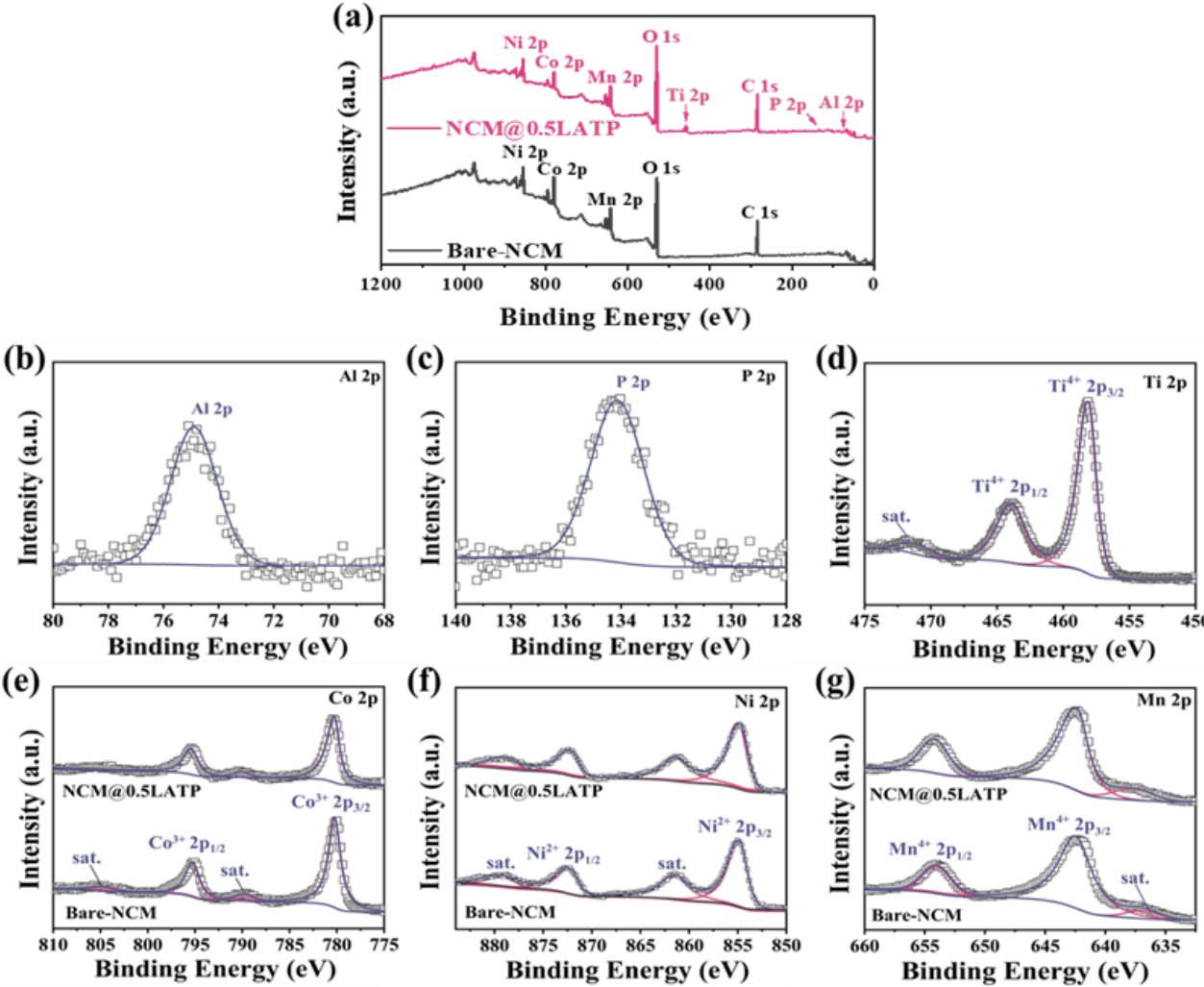

**Figure 5.** XPS spectra of (**a**) bare NCM and NCM@0.5LATP; high resolution spectra of (**b**) P 2p, (**c**) P 2p, (**d**) Ti 2p for NCM@0.5LATP, and (**e**) Co 2p, (**f**) Ni 2p, and (**g**) Mn 2p for bare NCM and NCM@0.5LATP.

Figure 6 shows the $N_2$ adsorption/desorption isotherms, and the shapes of the two samples are similar. However, for both samples, the end point of the desorption curve cannot match the start point of the adsorption curve. The reason may be the surface areas of two samples are relatively small [40,41]. Based on present data, the surface areas of bare NCM and NCM@0.5%LATP are calculated to be ~4.7661 $m^2$ $g^{-1}$ and ~5.7378 $m^2$ $g^{-1}$, respectively.

Galvanostatic charge–discharge and rate performances were tested for all three samples. For the bare NCM sample (Figure 7a), a significant declination in both voltage and capacity appears during the cycles. The reason is related to the HF corrosion of electrode materials by point decompression decomposition at high voltage [42]. The discharge capacity of bare NCM drops to 122.4 mAh $g^{-1}$ after 50 cycles at 0.2C, with a capacity retention of only 64.28%. The NCM@0.5LATP (Figure 7b) and NCM@1LATP (Figure 7c) samples share similar discharge curves and are both much better than the bare NCM. While the discharge capacity of NCM@0.5LATP is slightly larger than that of NCM@1LATP, which suggests the former sample presents a better performance at this rate. This result indicates that though LATP is a Li-ion conductor, a thick layer of LATP might decrease the conductivity of the electrode and degrade the discharge capacity [35,43–45]. Results in Figure 7e show that when the rate is increased to 1C, the discharge-specific capacities of all three sample materials decrease. The capacity decay of the bare NCM sample is more obvious than that

of the other samples. Figure 7f shows the rate performance of the three samples, and the chosen rates are 0.2C, 0.5C, 1C, 2C, and 5C, respectively. At small rates such as 0.2C, 0.5C, and 1C, the capacity of the bare NCM sample is larger than the other two samples, which is coincident with the cycle performances shown in Figure 7d,e. When the rate increases to 2C and 5C, the capacity of the bare NCM become much lower. At this time, the NCM@1LATP sample shows the largest capacity. It can still deliver a capacity of 70 mAh g$^{-1}$ at 5C. The above results indicate that NCM@0.5LATP exhibits the best cycling performance in the three samples at a cut-off voltage of 4.6 V. Earlier reports suggest some undesirable spinel phases grow during the cycles, and they will cause the decrease in voltage and capacity [46]. According to our results, the growth of spinel phase would be attenuated by the LATP layer, and the electrochemical performances are increased.

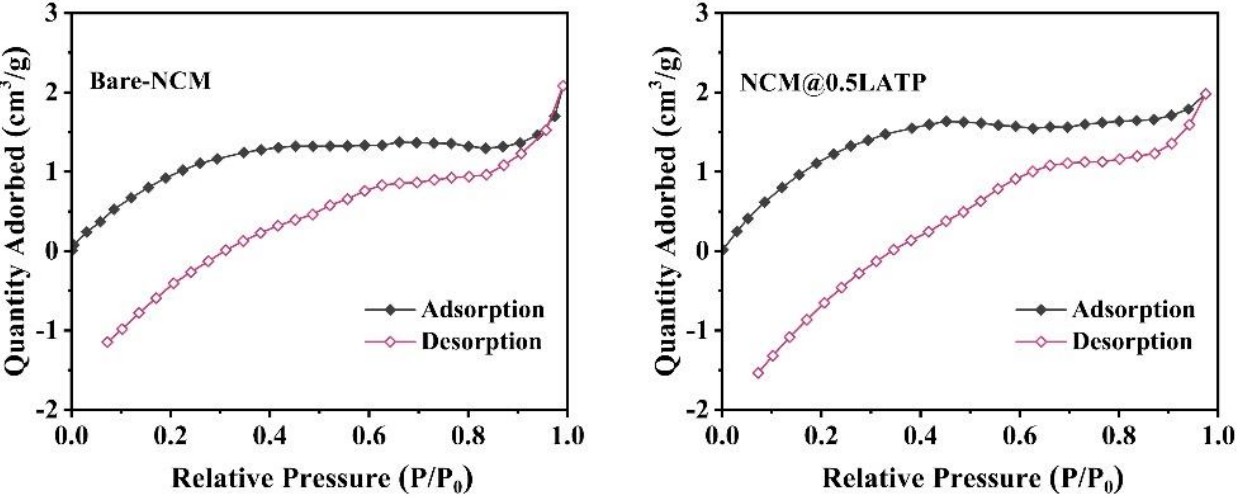

**Figure 6.** N$_2$ adsorption/desorption isotherms of bare NCM and NCM@0.5LATP.

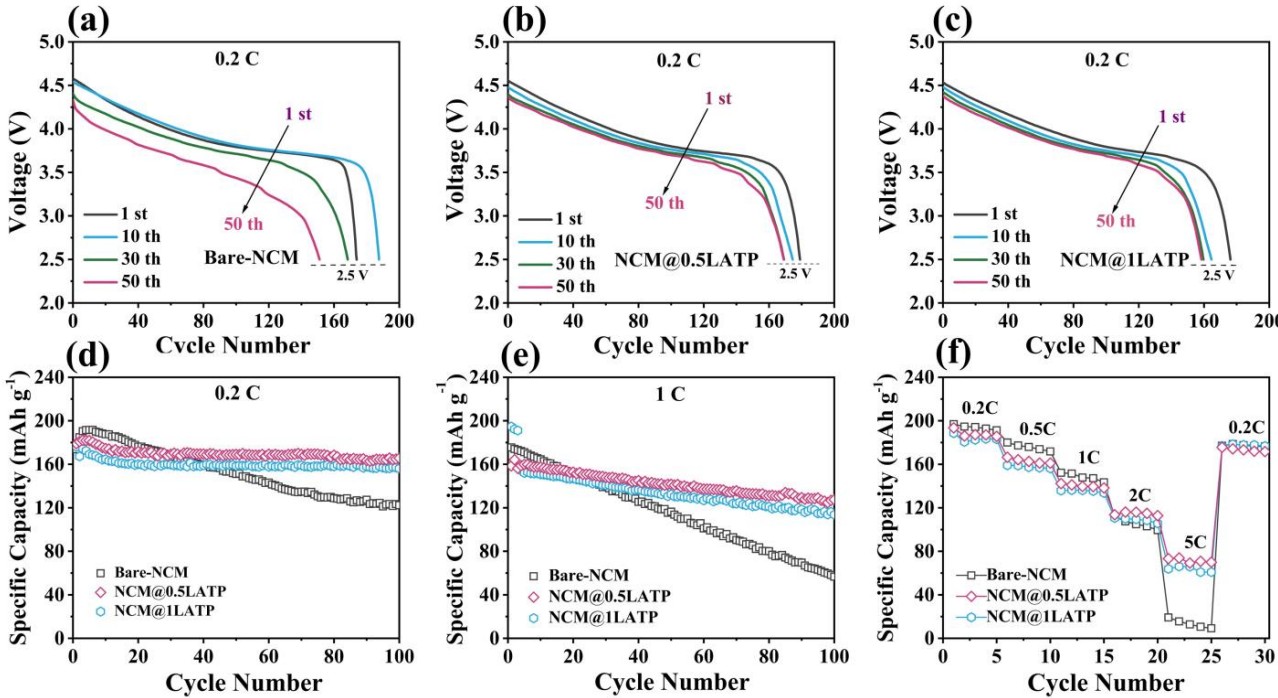

**Figure 7.** Discharge profiles of (**a**) bare NCM, (**b**) NCM@0.5LATP, and (**c**) NCM@1LATP; cycle performance of the three samples at (**d**) 0.2C and (**e**) 1C rate; (**f**) rate performances of the three samples.

To reveal the effect of the LATP layer on the electrochemical internal resistances, EIS experiments were performed on bare NCM and NCM@0.5LATP (Figure 8). The Nyquist curves of the two samples are similar. The equivalent circuit model used for fitting is shown in the inset. R1 in the model represents the Ohmic resistance, which may mainly come from the electrolyte, while R2 represents the charge transfer resistance. Table 1 shows the fitted parameters. According to the data, R2 of the bare NCM is 93.1 $\Omega$ before cycle, and increases to 146.7 $\Omega$ after 50 cycles, with a growth rate of 57.6%. While for NCM@0.5LATP, R2 increases from 59.6 $\Omega$ to 75.3 $\Omega$, with a growth rate of only 26.3%. This result indicates the surface layer of NCM@0.5LATP is much stable than that of the bare NCM. The reason may be the protective layer prevents the side reactions of the electrolyte from occurring, and forms a stable cathode–electrolyte interphase, which provides a better microporous channel for lithium-ion diffusion [13,14].

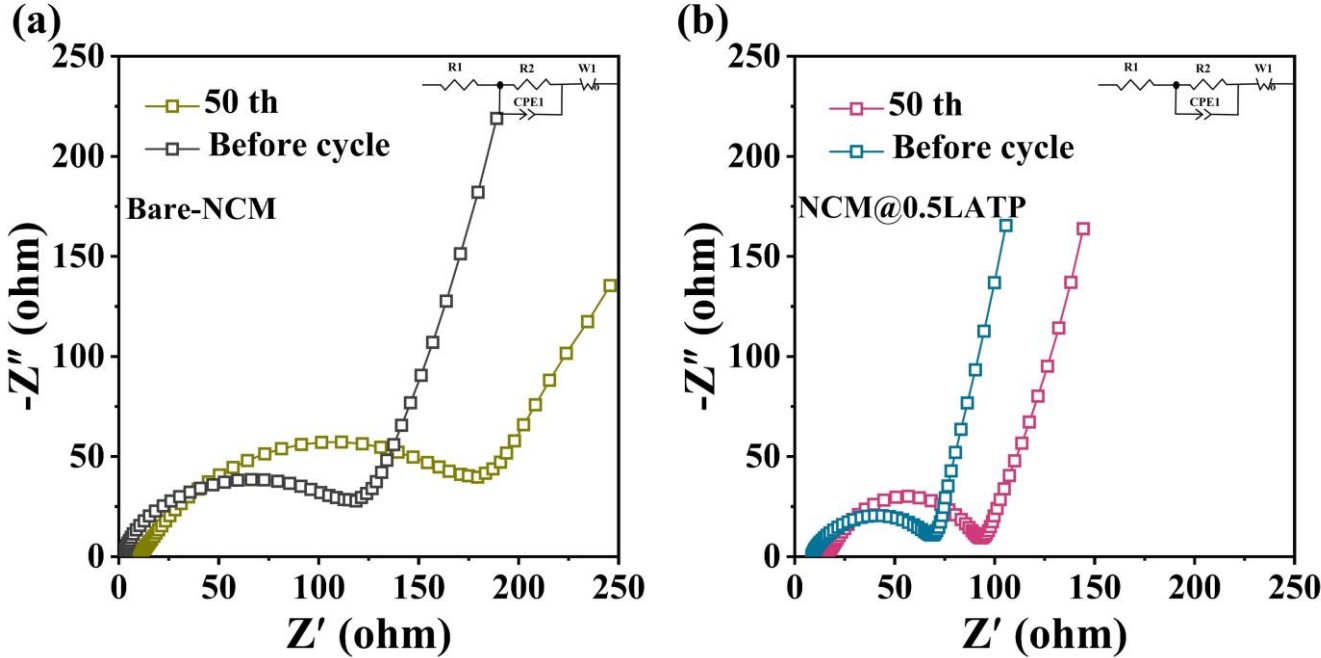

**Figure 8.** EIS results of (**a**) bare NCM and (**b**) NCM@0.5LATP; inset is the used equivalent circuit model.

**Table 1.** Fitted impedance parameters.

| Cycle Number | Before Cycle R$_2$ ($\Omega$) | 50th Cycle R$_2$ ($\Omega$) |
|---|---|---|
| NCM@0.5LATP | 59.6 | 75.3 |
| Bare-NCM | 93.1 | 146.7 |

## 4. Conclusions

In summary, a homogeneous LATP layer was prepared on an NCM surface by a simple wet chemical method. The sample with 0.5 wt% LATP presents an increased capacity retention of 92.17% after 100 cycles at 0.2C. It also exhibits a better rate performance than the other two samples. EIS result indicates the surface layer of NCM@0.5LATP is much stable than that of the bare NCM. Our results suggest that LATP-surface coating might be an effective method to improve the cycle stability of NCM, and it might also be useful for other high-capacity cathode materials.

**Author Contributions:** Data curation, B.H.; Formal analysis, Y.G.; Investigation, M.Z.; Methodology, W.W.; Project administration, J.J.; Supervision, H.W.; Writing—original draft, P.Z.; Writing—review & editing, R.W. All authors have read and agreed to the published version of the manuscript.

**Funding:** This research was funded by Research on high power flexible battery in all sea depth (2020-XXXX-XX-246-00).

**Institutional Review Board Statement:** Not applicable.

**Informed Consent Statement:** Not applicable.

**Conflicts of Interest:** The authors declare no conflict of interests.

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
