# Peer review of "Enhanced Electrochemical Performance of LiNi1/3Co1/3Mn1/3O2 at a High Cut-Off Voltage of 4.6 V by Li1.3Al0.3Ti1.7(PO4)3 Coating"

_coatings, doi:10.3390/coatings12121964_

Round 1

Reviewer 1 Report

In this manuscript, the authors reported an improved electrochemical performance of LiNi1/3Co1/3Mn1/3O2 by coating of Li1.3Al0.3Ti1.7(PO4)3 (LATP). It is an interesting topic in the field of energy storage and conversion. However, major revision as follow should be clarified/conducted before it can be published.

In the experimental section the authors mention, "Commercial LiNi1/3Co1/3Mn1/3O2 (NMC)…”. Author should correct NMC as NCM.

Author should include the BET analysis of the cathode materials. Because, the electrochemical performance of electrode depends on surface area of the active materials.

The formation and presence of Li1.3Al0.3Ti1.7(PO4)3 (LATP) over the surface of  LiNi1/3Co1/3Mn1/3O2 (NCM) is not properly explained.

In tha XRD and XPS analysis, there was no change was found in the XRD pattern after coating; there was no Al and P signals were found in the XPS analysis. Then how authors confirms the formation of LATP?

The authors claimed that the presence of Li1.3Al0.3Ti1.7(PO4)3 (LATP) was confirmed by SEM-EDX and TEM analysis. However, it is too difficult to agree the result presented in the SEM images, because there was no change observed on the LiNi1/3Co1/3Mn1/3O2 (NCM) surface.    

Check Figure 5 (a) and correct it.

Why electrochemical performance of 1%LATP slight lower than 0.5%LATP?  Author should include more information with adequate ref, because thickness of coating was properly explained.

Reviewer 2 Report

Through the review, I think this is a very nicely written paper, and the author analyzed and discussed the results in detail and correctly. The analysis developed in this paper is correct and the obtained results are interesting. The paper has sufficient novelty and covers the scope of the Coatings journal. Therefore, the manuscript may consider for publication in the Coatings journal after responding to the following comments and revising the manuscript properly.

1. The novelty of the work is missing in the introduction section. Explain it properly.

2. Improve language throughout the manuscript.

3. Reduce similarity. Check the attached Turnitin Report.

4. Improve the introduction based on the following reference on electrochemical energy devices: (Ceramics International Volume 47, Issue 17, 1 September 2021, Pages 23725-23748; Nanomaterials 2022, 12(20), 3581; Ceramics International Volume 48, Issue 22, 15 November 2022, Pages 32588-32612; ACS Appl. Electron. Mater. 2022, 4, 7, 3327–3353; )

Reviewer 3 Report

The manuscript presents an investigation on improving the long-term cycling performance of LiNi1/3Co1/3Mn1/3O2 (NCM) through surface coating of Li1.3Al0.3Ti1.7(PO4)3 (LATP) at a high cut-off voltage of 4.6V. After 100 cycles at 0.2C (1C = 200mA g-1), the battery prepared with 0.5 wt% LATP-modified NCM exhibits increased discharge capacity retention of 92.37%, while the bare NCM only presents 64.28%.

Although the study is quite significant, however very similar study have already been reported previously by Yi Wang et al, 2019 Chinese Phys. B 28 068202. They have reported that 1 wt% Li1.4Al0.4Ti1.6(PO4)3 (LATP) modified Li(Ni0.6Co0.2Mn0.2)O2 (NCM622) exhibits the best cyclic stability at high charging cut-off voltage of 4.5 V. After 100 cycles at 1 C rate (1 C = 190 mA g-1), it displayed capacity retention of 90.9%, then that of bare material (79%).

Can authors elaborate what are the basic difference in this study and how this study is significant. From a novelty perspective, the manuscript can only be ready for publication in its current form after a comprehensive assessment of the importance of this study.

Round 2

Reviewer 1 Report

Authors have clarified all the comments. However, author should include BET results in the manuscript with proper explanation. The manuscript can be accepted after inclusion of BET results. 
